# Lipid/Clay-Based Solid Dispersion Formulation for Improving the Oral Bioavailability of Curcumin

**DOI:** 10.3390/pharmaceutics14112269

**Published:** 2022-10-24

**Authors:** Jae Geun Song, Hye-Mi Noh, Sang Hoon Lee, Hyo-Kyung Han

**Affiliations:** BK21 FOUR Team and Integrated Research Institute for Drug Development, College of Pharmacy, Dongguk University-Seoul, Dongguk-ro-32, Ilsan-Donggu, Goyang 10326, Korea

**Keywords:** curcumin, krill oil, lipid-based formulation, aminoclay, antisolvent precipitation

## Abstract

This study was conducted to develop a lipid/clay-based solid dispersion (LSD) formulation to enhance the dissolution and oral bioavailability of poorly soluble curcumin. Krill oil and aminoclay were used as a lipid and a stabilizer, respectively, and LSD formulations of curcumin were prepared by an antisolvent precipitation method combined with freeze-drying process. Based on the dissolution profiles, the optimal composition of LSD was determined at the weight ratio of curcumin: krill oil: aminoclay of 1:5:5 in the presence of 0.5% of D-α-tocopherol polyethylene glycol succinate. The structural and morphological characteristics of the LSD formulation were determined using X-ray powder diffraction, differential scanning calorimetry, and scanning electron microscopy. Crystalline curcumin was changed to an amorphous form in the LSD formulation. At the pH of acidic to neutral, the LSD formulation showed almost complete drug dissolution (>90%) within 1 h, while pure curcumin exhibited minimal dissolution of less than 10%. Furthermore, the LSD formulation had significantly improved oral absorption of curcumin in rats, where C_max_ and AUC of curcumin were 13- and 23-fold higher for the LSD formulation than for the pure drug. Taken together, these findings suggest that the krill oil-based solid dispersion formulation of curcumin effectively improves the dissolution and oral bioavailability of curcumin.

## 1. Introduction

Curcumin is a polyphenolic flavonoid extracted from the rhizomes of *Curcuma longa* [1]. It exhibits therapeutic benefits in preventing some diseases including cancer, obesity, infectious and cardiovascular diseases. Curcumin has a favorable safety profile with low toxicity [2]. Although curcumin has diverse biological effects and a good safety profile, its clinical application is limited because of low aqueous solubility and poor bioavailability [3]. Therefore, effective solubilization of curcumin is critical in improving its oral bioavailability and expanding its therapeutic applications.

Various formulation approaches, including nanoparticles, solid dispersions, and liposomes, have been pursued to improve the oral bioavailability of curcumin [4,5,6,7,8,9]. The lipid-based formulations are promising in enhancing the oral bioavailability of poorly soluble drugs, where lipophilic drugs are solubilized in lipids and may undergo particle size reduction via antisolvent precipitation [10]. In addition to particle size reduction, a lipid-based formulation can bypass the hepatic first-pass effect via the lymphatic absorption pathway, leading to enhancing oral absorption [11,12,13,14]. Among various lipids, krill oil, rich in phospholipids coupled with long-chain omega-3 polyunsaturated fatty acids, is useful for the preparation of lipid-based formulations [15]. Krill oil has some advantages, such as (i) phospholipid components can effectively solubilize lipophilic drugs due to their amphiphilic property [16], (ii) omega-3-coupled phospholipids can release docosahexaenoic acid (DHA) and eicosapentaenoic acid (EPA) after oral absorption, providing additional health-promoting effect [17], and (iii) it contains various potent antioxidants (e.g., provitamin E, flavonoids, and vitamin A) [18]. Therefore, in the present study, krill oil was selected as a lipid component, and the lipid-based solid dispersion of curcumin was prepared using an antisolvent precipitation method coupled with a subsequent freeze-drying process to prevent particle growth [19]. Given that 3-aminopropyl-functionalized magnesium phyllosilicate (aminoclay) is an effective cryoprotectant, superior to commonly used sucrose, it was also added to lipid-based solid dispersions as a stabilizer [20]. Since aminoclay is dispersed in water as a cationic nanosheet and krill oil has a negative charge, the krill oil-based solid dispersion may be further stabilized by aminoclay to prevent aggregation during freeze-drying [21]. Surfactants improve the drug solubility and stabilize the amorphous solid dispersions by avoiding recrystallization [22]. D-a-tocopheryl polyethylene glycol 1000 succinate (TPGS) is a non-ionic surfactant that can enhance drug solubility and stability [23]. Therefore, in the present study, lipid/clay-based solid dispersion (LSD) formulations of curcumin were prepared with krill oil, aminoclay, and TPGS and their in vitro/in vivo characteristics were evaluated.

## 2. Materials and Methods

### 2.1. Materials

Curcumin, D-α-tocopheryl polyethylene glycol 1000 succinate, Tween^®^ 80, and propionic acid were purchased from Sigma Co. (St Louis, MO, USA). Krill oil and sodium taurocholate were purchased from Shanghai ColdSpring Biopharma Co., Ltd. (Shanghai, China). Low-substituted hydroxypropyl cellulose (L-HPC) was obtained from Shin-Etsu Chemical Co., Ltd. (Tokyo, Japan). Lecithin and acetone were purchased from Daejung chemical & metals Co., Ltd. (Gyeonggi-do, Korea). Acetonitrile and methanol were provided by Avantor, Inc. (Radnor, PA, USA). Aminoclay was obtained using the method reported previously [24]. All other chemicals were of analytical grade and all solvents were of HPLC grade.

### 2.2. Preparation of Lipid/Clay-Based Solid Dispersion

LSD formulations were prepared by the antisolvent precipitation method followed by an immediate freeze-drying process. First, curcumin and krill oil were dissolved in acetone, while TPGS and aminoclay were dissolved in water. The drug and excipient ratios varied, as shown in Table 1. These two solutions (aqueous and organic) were mixed under vigorous stirring and underwent freeze-drying immediately after mixing. The mixture was rapidly frozen at −80 °C and lyophilized using a lyophilizer (LYOPH-PRIDE 10R, Ilshin BioBase Co. Ltd., Gyeonggi-do, South Korea). Lyophilization was conducted at a shelf temperature of −35 °C and a condenser temperature of −60 °C for 24 h. The shelf temperature was increased gradually to 20 °C over 24 h.

### 2.3. Structural and Morphological Characterization

The differential scanning calorimetry (DSC) analysis was performed using a DSC Q2000 (TA Instruments, Ghent, Belgium). The samples (5–10 mg) were placed into a hermetically sealed aluminum pan. The thermograms were obtained from 20 °C to 200 °C at the scanning rate of 10 °C/min in an inert atmosphere flushed with nitrogen at a rate of 50 mL/min.

Drug crystallinity was examined by X-ray powder diffraction (XRPD) analysis using an X-ray diffractometer (X’Pert APD; PHILIPS, Amsterdam, the Netherlands) with CuKα radiation at 20 mA and 40 kV. Each sample was mounted on a glass X-ray sample holder. Scans were recorded between 2° and 70° (2*θ*) with a step size of 0.04° and a scanning speed of 3°/min. XRPD experiments were conducted at Korea Basic Science Institute (Daegu Center, Daegu, South Korea).

The morphological characteristics of pure curcumin and LSD formulations were determined using scanning electron microscopy (SEM). Each formulation was spread on a specimen stub using double-sided sticky tape, coated with platinum, and analyzed by a field-emission scanning electron microscope (S-3000N; Hitachi, Tokyo, Japan).

The particle size of the LSD formulation was measured by dynamic light scattering (DLS) using a Zetasizer Nano-ZS90 (Malvern Instruments, Malvern, UK). The polydispersity index (PDI) was also obtained to assess the size distribution. The measurements were carried out in triplicates and performed at 25 °C.

### 2.4. Solubility Study

The aqueous solubility of curcumin in each formulation was assessed by the shake-flask method. An excess amount of each formulation (equivalent to 3 mg of curcumin) was added into water (5 mL), and then the suspension was equilibrated for 48 h under stirring (200 rpm) at 25 °C. Samples were filtered through PVDF syringe filters (pore size: 0.45 μm, diameter: 15 mm). Drug concentration in the filtrate was analyzed using an Ultra-performance liquid chromatography (UPLC) assay.

### 2.5. In Vitro Drug Release Study

Dissolution studies were conducted using the USP paddle method in the dissolution tester DT 1420 (ERWEKA, Heusenstamm, Germany) at 50 rpm and 37 °C. Each formulation (equivalent to 1 mg of curcumin) was filled in hard gelatin capsules with 5% L-HPC, which was then added into dissolution media (500 mL) with 2% tween-80. Samples were collected at the predetermined time points, and then the equal volume of fresh medium was replenished into the vessel to maintain the constant volume of dissolution media. The collected samples were filtered through PVDF syringe filters (pore size: 0.45 μm, diameter: 15 mm). The released drug amount was analyzed by a UPLC assay.

Dissolution characteristics were also examined at acidic to neutral pH ranges (pH 1.2, 4.0, and 6.8). Each formulation (equivalent to 10 mg of curcumin) was filled into hard gelatin capsules with 5% L-HPC and added into dissolution media (900 mL) containing 2% tween-80. Samples were collected at the predetermined time points (10, 15, 30, 45, 60, 90, and 120 min) and filtered through PVDF syringe filters (pore size: 0.45 μm, diameter: 15 mm). Fresh medium was replenished to the vessel to maintain the constant volume of dissolution media. The released drug amount was analyzed by a UPLC assay.

### 2.6. Dissolution Studies in Simulated Intestinal Fluids

Drug release profiles of the LSD formulation were examined in fasted-state simulated intestinal fluid (FaSSIF) and fed-state simulated intestinal fluid (FeSSIF) using the USP paddle method as described above. The FaSSIF and FeSSIF were prepared following the previously reported method [25,26,27]. FaSSIF was composed of sodium taurocholate (0.003 mmol/mL), lecithin (0.0002 mmol/mL), maleic acid (0.01912 mmol/mL), sodium hydroxide (0.0348 mmol/mL), and sodium chloride (0.06862 mmol/mL). The composition of FeSSIF included sodium taurocholate (0.01 mmol/mL), lecithin (0.002 mmol/mL), maleic acid (0.05502 mmol/mL), sodium hydroxide (0.08165 mmol/mL), sodium chloride (0.1255 mmol/mL), glyceryl monooleate (0.005 mmol/mL), and sodium oleate (0.0008 mmol/mL). The pH of FaSSIF and FeSSIF was adjusted to 6.5 and 5.8, respectively. Dissolution studies were performed at 50 rpm in 900 mL of dissolution media. Samples were collected at predetermined time points (10, 15, 30, 45, 60, 90, and 120 min) and were filtered using PVDF syringe filters (pore size: 0.45 μm, diameter: 15 mm). Fresh medium was replenished to the vessel to maintain the constant volume of dissolution media. Drug concentration in the filtrate was assessed by a UPLC assay.

### 2.7. Storage Stability

Stability of the developed LSD formulation was evaluated during the storage for 2 months. The LSD formulation (powder) was placed into an amber vial and stored at 4 °C and 25 °C for 2 months. At the predetermined time points, samples were collected to examine their dissolution and morphological characteristics. Alteration in drug crystallinity was also evaluated by XRPD analysis.

### 2.8. Pharmacokinetic Studies

Male Sprague-Dawley rats (250–280 g) were supplied by Orientbio (Seongnam, South Korea). The experimental protocol was approved by the review committee of Dongguk University (IACUC-2021-061). Before the experiments, rats were fasted for 12 h and divided into two groups (n = 6 per group). Rats were administered pure curcumin (group 1) or LSD-F3 (group 2) orally at a dose equivalent to 100 mg/kg of curcumin. Each formulation was dispersed in 0.5% aqueous methylcellulose. Blood samples were collected from the femoral artery at 0.25, 0.5, 0.75, 1, 2, 4, 6, and 8 h after the dosing. After centrifuging blood samples at 16,600× *g* for 5 min at 4 °C, the obtained plasma was stored at −20 °C until analyzed.

### 2.9. Analytical Assay

In vitro samples: Drug concentration was determined by a UPLC assay using a Waters^®^ ACQUITY UPLC^®^ System (Hertfordshire, UK) and a reverse phase C18 column (Kinetex^®^, C18, 2.1 × 100 mm, 1.7 µm; Phenomenex, Torrance, CA, USA). The mobile phase consisted of 0.1% propionic acid in acetonitrile: 0.1% propionic acid in water (65:35, *v*/*v*). The flow rate was 0.2 mL/min, and the detection wavelength was 425 nm. The column temperature was 35 °C. Sudan Ⅰ was used as an internal standard (IS), and the calibration curve of curcumin was linear over the concentration range of 0.25–10 µg/mL with good linearity (*r*^2^ > 0.999).

In vivo samples: To determine the drug concentration in plasma, 100 μL of each plasma sample was mixed vigorously with 20 μL of Sudan Ⅰ (IS, 10 µg/mL). Methanol (180 μL) was added and vortexed vigorously, and then the mixture was centrifuged at 16,600× *g* for 10 min. The supernatant was dried under a vacuum. The residue was reconstituted with the mobile phase and injected into the UPLC system as described above. The mobile phase was composed of 0.1% propionic acid in acetonitrile: 0.1% propionic acid in water (65:35, *v*/*v*) with the flow rate of 0.2 mL/min at 35 °C. The detection wavelength was 425 nm, and the calibration curve was linear over the concentration range of 0.01–0.5 µg/mL (*r*^2^ > 0.999).

### 2.10. Pharmacokinetic and Statistical Analysis

The area under the plasma concentration-time curve (AUC) was calculated using the linear trapezoidal method. The maximum plasma concentration (C_max_) and the time to reach the maximum plasma concentration (T_max_) were determined directly from the visual inspection of plasma concentration-time data.

The data are represented as mean values with standard deviation. Statistical analysis was performed using a student *t*-test or one-way ANOVA followed by Dunnett’s test. A *p*-value less than 0.05 was considered a statistically significant difference.

## 3. Results and Discussion

### 3.1. Optimization of LSD Formulations

Single-factor analysis based on drug release profiles was used to optimize the composition of LSD formulations. The formulation variables and compositions are summarized in Table 1.

The effect of each formulation component on drug dissolution was examined as summarized in Figure 1. Since Omega-3 phospholipids in krill oil could serve as a surfactant, the rate and extent of drug dissolution tend to increase as the proportion of krill oil in LSD formulations increased (Figure 1A). Amphiphilic properties of krill oil contributed to solubilizing curcumin by lowering the interfacial tension and improving the wettability of drug particles. In addition, krill oil could be adsorbed on the surface of drug particles via the hydrophobic interaction between omega-3 fatty acids and nonpolar drugs, providing a mechanical barrier to coalescence and improving the thermodynamic stability of the LSD formulations [28,29,30]. This results in the inhibition of particle growth, leading to the enhanced drug dissolution. Therefore, the weight ratio of curcumin to krill oil was selected as 1:5 for subsequent studies.

The effect of aminoclay on the drug release from LSD formulations was also examined with the varying drug:aminoclay ratios. As shown in Figure 1B, the dissolution of curcumin was enhanced by increasing the weight ratio of aminoclay. In particular, the LSD formulation having a drug:aminoclay ratio of 1:5 exhibited rapid drug release and achieved approximately 94% of drug release within 15 min, while the other formulations displayed less than 80% of drug dissolution. Considering the effect of aminoclay as a cryoprotectant, drug particles dispersed in krill oil could be further stabilized in the solid matrix of aminoclay to prevent aggregation during freeze-drying [21,31,32]. As a result, the higher proportion of aminoclay might minimize particle growth and aggregation, resulting in enhanced drug dissolution. Therefore, the weight ratio of curcumin to aminoclay was set at 1:5.

Since surfactants improve the wettability and physical stability of drugs, the effect of TPGS on the drug release from LSD formulations was also examined (Figure 1C). As an amphiphilic and water-soluble derivative of natural vitamin E, it has been actively applied to enhance the aqueous solubility of lipophilic drugs. In addition, TPGS can enhance the cellular uptake of P-gp substrates [33], exerting a synergistic effect with curcumin in P-gp inhibition [4]. When the proportion of TPGS increased from 0.1% to 0.5%, drug dissolution increased from 72% to 94%. This may be because amphiphilic TPGS lowered interfacial tension and enhanced the wettability of drug particles [34]. Since LSD formulation achieved almost complete drug dissolution in the presence of 0.5% TPGS, a further increase in TPGS proportion from 0.5% to 1.0% did not show any statistically significant difference in drug dissolution. Therefore, 0.5% TPGS was used in the final formulation.

Among the tested formulations, the F3 formulation (LSD-F3) at the weight ratio of drug: krill oil:aminoclay of 1:5:5 in the presence of 0.5% TPGS showed rapid and extensive drug dissolution. Thus, LSD-F3 was selected as an optimal formulation for curcumin, and further characterization proceeded with the LSD-F3 formulation.

### 3.2. Characterization of the Optimized LSD Formulation

#### 3.2.1. Structural and Morphological Characteristics

The crystalline state of curcumin in LSD formulation was examined by DSC and XRPD analysis. As shown in Figure 2, while a sharp endothermic peak was observed at the melting point of 180 °C for pure drug powder, it was not observed in the DSC curve of LSD-F3 formulation. Since the disappearance of endothermic drug peak may be due to the solubilization of drugs in the melted matrix other than the change in drug crystallinity, XRPD analysis was performed to confirm the drug crystalline state. As shown in Figure 3, pure drug powder displayed many distinct peaks, indicating a highly crystalline state of curcumin [35]. On the other hand, these distinct diffraction peaks of curcumin were absent in the XRPD diffractogram of LSD-F3, indicating that the drug dispersed in the lipid-aminoclay matrix might be in an amorphous form.

The morphological characteristics of pure curcumin and LSD-F3 were also investigated by SEM. As shown in Figure 4, pure curcumin showed a flat, rod-like crystal structure, while LSD-F3 exhibited a homogeneous blend of drug and excipients in the form of coarse and irregular-shaped particles. This result is also consistent with the observations from DSC and XRPD analysis, suggesting the amorphous state of the drug in LSD-F3.

#### 3.2.2. Solubility

The aqueous drug solubility in LSD-F3 formulation was compared to pure drug powder. While pure drug powder exhibited very low aqueous solubility of 0.05 µg/mL, LSD-F3 formulation significantly enhanced the aqueous solubility of curcumin to 19.66 ± 0.46 µg/mL, which was about 400-fold higher than that of pure drug powder. In general, the improved aqueous solubility of hydrophobic drugs in solid dispersion depends on (i) drug dispersion in amorphous form, (ii) inhibition of drug particle growth, and (iii) capability to increase the saturated solubility in an aqueous state [36,37,38]. Similarly, the enhanced drug solubility via LSD-F3 formulation can be explained by multiple factors. Drug crystallinity was changed to an amorphous state in LSD-F3 (Figure 2, Figure 3 and Figure 4), leading to enhanced drug solubility. In addition, phospholipids in krill oil and aminoclay stabilized drug particles, minimizing particle growth during antisolvent precipitation and freeze-drying process as described in Section 3.1. TPGS in LSD-F3 could also contribute to solubilizing curcumin via lowering interfacial tension and forming micelles. Overall, LSD-F3 was effective at improving the aqueous solubility of curcumin.

#### 3.2.3. Dissolution Characteristics

The dissolution behavior of LSD-F3 formulation was examined over the pH range of acidic to neutral and compared to that of pure drug powder. While pure drug powder exhibited minimal dissolution of less than 10% (Figure 5A), LSD-F3 dramatically increased the rate and the extent of drug release at all the tested pH, indicating pH-independent drug dissolution (Figure 5B). LSD-F3 achieved almost complete dissolution (>90%) within 1 h at the pH range of 1.2–6.8, suggesting an efficient drug release along the GI tract. These results may be explained by the generation of fine drug particles via the antisolvent precipitation method followed by immediate freeze-drying. As dispersed in water, LSD-F3 showed an average particle size of 204.8 ± 2.08 nm with PDI values of 0.226 ± 0.005, indicating the formation of fine drug particles. During antisolvent precipitation, phospholipids in krill oil served as a short-term stabilizer to retard or prevent particle growth by surrounding the surface of drug particles. In addition, aminoclay acting as a cryoprotectant may inhibit the aggregation and growth of drug particles during the freeze-drying process [21,31,32,39]. Particle size reduction increased the surface area of particles, leading to enhanced drug dissolution. The change in drug crystallinity to an amorphous form could enhance drug dissolution. The improved surface-wetting by phospholipids in krill oil and TPGS may also enhance drug dissolution [16,34].

Given that food intake may affect the drug release from LSD formulations, dissolution profiles of LSD-F3 were evaluated in simulated intestinal fluids (FaSSIF and FeSSIF) to mimic the fasted and fed conditions. As summarized in Figure 6, the dissolution of the pure drug was significantly enhanced in the fed condition compared to that in the fasted condition. Some factors may explain this; first, to reflect the biliary response to food intake, the concentrations of bile salts and phospholipids were considerably higher in FeSSIF than in FaSSIF [25,40]. Since bile salts and phospholipids can facilitate the wetting of drug particles and the solubilization of hydrophobic drugs into mixed micelles [41,42], they could enhance the drug dissolution to a greater extent in FeSSIF. Furthermore, lipid digestion products (glyceryl monooleate and sodium oleate) in FeSSIF could also increase the solubility of hydrophobic drugs [41]. Consequently, pure drug powder exhibited greater drug dissolution in FeSSIF than in FaSSIF (Figure 6). Conversely, LSD-F3 achieved rapid and almost complete drug release in FeSSIF and FaSSIF, implying that LSD-F3 may minimize the effect of food intake on drug absorption.

#### 3.2.4. Storage Stability

Amorphous solid dispersion may undergo recrystallization during storage, resulting in a change in drug dissolution profiles. Therefore, the drug dissolution behavior of LSD-F3 was examined during the storage at 4 °C and 25 °C. As summarized in Table 2, the dissolution characteristics of LSD-F3 were not altered after 2 month-storage at 4 °C and 25 °C, displaying a similarly high (>90%) drug dissolution. In addition, there was no change in morphological characteristics and amorphous state of LSD-F3 after 2 month-storage (Figure 3 and Figure 4).

#### 3.2.5. Pharmacokinetics

The plasma concentration-time profiles of curcumin after an oral dose of pure curcumin and LSD-F3 are illustrated in Figure 7, and pharmacokinetic parameters are summarized in Table 3. As shown in Figure 7, LSD-F3 significantly improved the oral exposure of curcumin compared to pure drug powder. LSD-F3 achieved the C_max_ and AUC 13- and 23-fold higher, respectively, than those from pure drug powder. In addition, LSD-F3 showed short T_max_, implying fast drug absorption. These results are consistent with the observation from in vitro dissolution studies. The improved oral bioavailability of curcumin via LSD-F3 formulation may be explained by multiple factors. First, given that the oral absorption of curcumin is solubility- and dissolution rate-limited, the enhanced aqueous solubility and dissolution of curcumin via LSD-F3 promoted intestinal drug absorption. Second, phospholipid components and TPGS in LSD-F3 formulation could form the micelles and encapsulate drugs into micellar cores, facilitating the drug diffusion across the intestinal membrane [43,44]. Third, the lipid-based formulation can bypass the first-pass effect via the lymphatic absorption pathway [11,12,45,46]. Since curcumin undergoes extensive hepatic metabolism [47], bypassing the first-pass effect via LSD-F3 could improve the oral bioavailability of curcumin. Collectively, LSD-F3 was effective at improving the oral absorption of curcumin in rats.

## 4. Conclusions

Lipid/clay-based solid dispersion formulation of curcumin (LSD-F3) was prepared by the antisolvent precipitation method followed by an immediate freeze-drying process. The optimal composition of LSD-F3 was determined as the weight ratio of curcumin: krill oil: aminoclay of 1:5:5 in the presence of 0.5% TPGS. The developed LSD-F3 effectively enhanced the rate and extent of drug dissolution, improving the oral exposure of poorly soluble curcumin in rats.

## Figures and Tables

**Figure 1 pharmaceutics-14-02269-f001:**
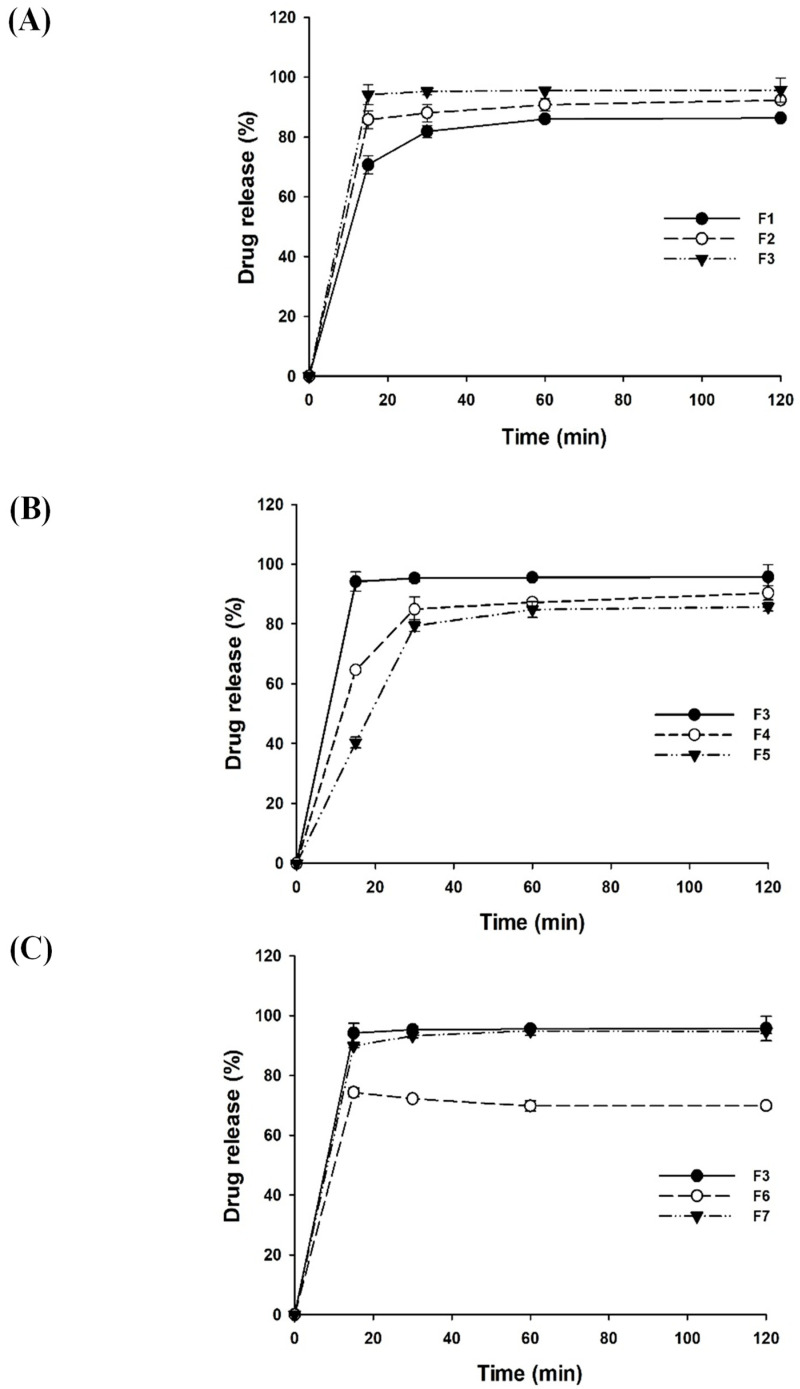
Effect of krill oil (**A**), aminoclay (**B**), and TPGS (**C**) on drug release (Mean ± SD, n = 3).

**Figure 2 pharmaceutics-14-02269-f002:**
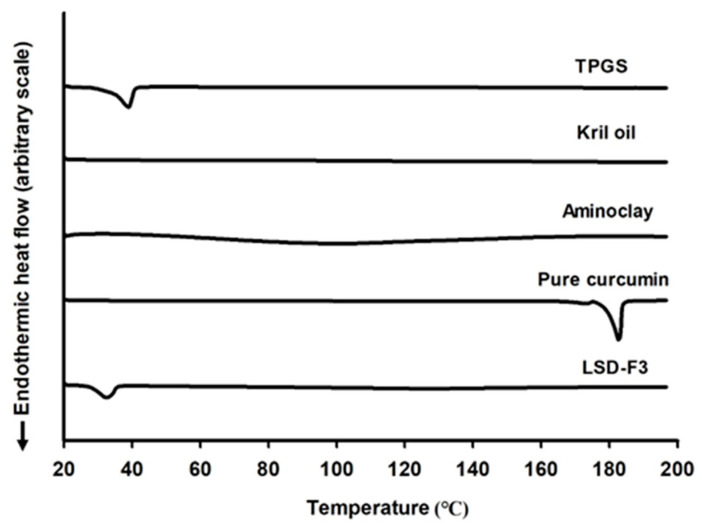
DSC curves of pure curcumin, LSD-F3, and formulation components.

**Figure 3 pharmaceutics-14-02269-f003:**
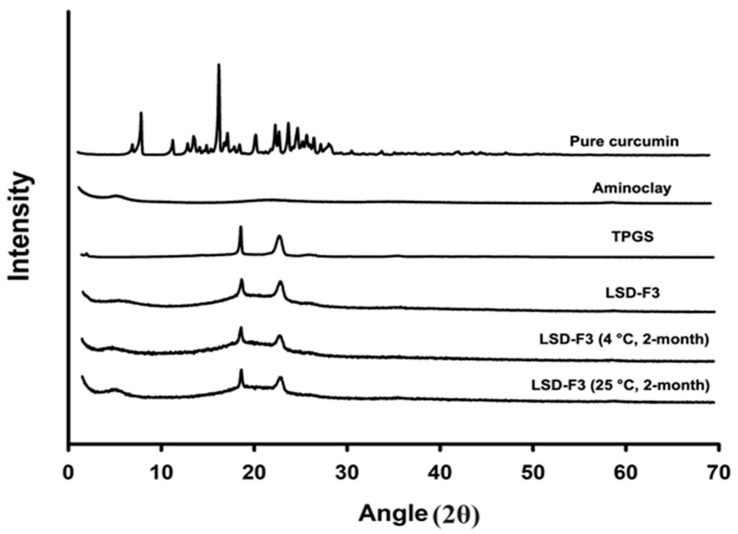
XRPD diffractograms of pure curcumin, formulation components, and LSD-F3 with/without 2-month storage at different temperatures.

**Figure 4 pharmaceutics-14-02269-f004:**
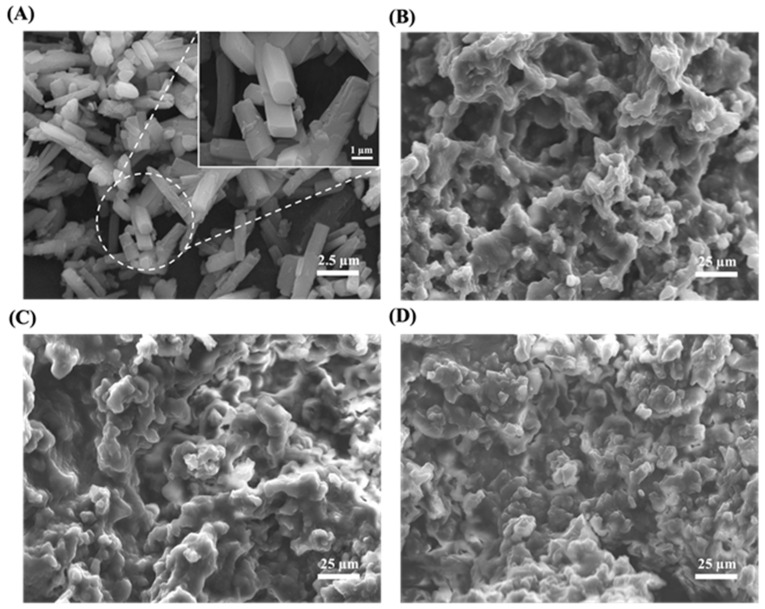
SEM images of pure curcumin and LSD-F3. (**A**): pure curcumin, (**B**): LSD-F3 at Day 0, (**C**) LSD-F3 after 2 month-storage at 4 °C, and (**D**) LSD-F3 after 2 month-storage at 25 °C.

**Figure 5 pharmaceutics-14-02269-f005:**
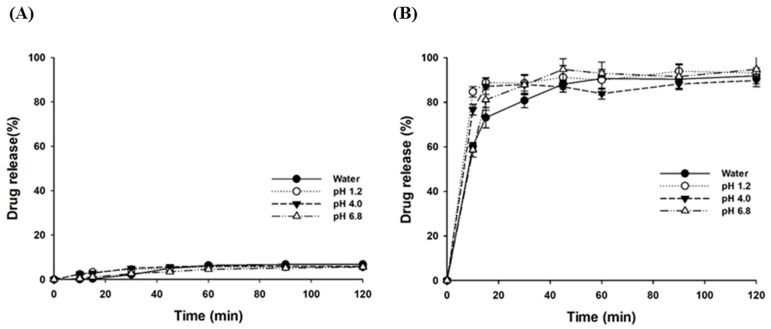
Dissolution profiles of pure curcumin (**A**) and LSD-F3 (**B**) at different pH values (Mean ± SD, n = 3).

**Figure 6 pharmaceutics-14-02269-f006:**
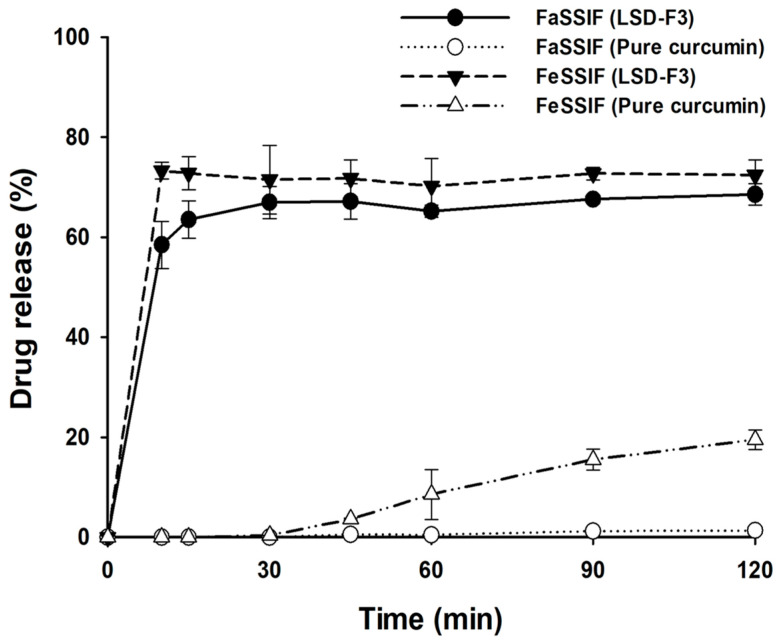
Drug release profiles of LSD-F3 in fasted-state simulated intestinal fluid (FaSSIF) and fed-state simulated intestinal fluid (FeSSIF) (Mean ± SD, n = 3).

**Figure 7 pharmaceutics-14-02269-f007:**
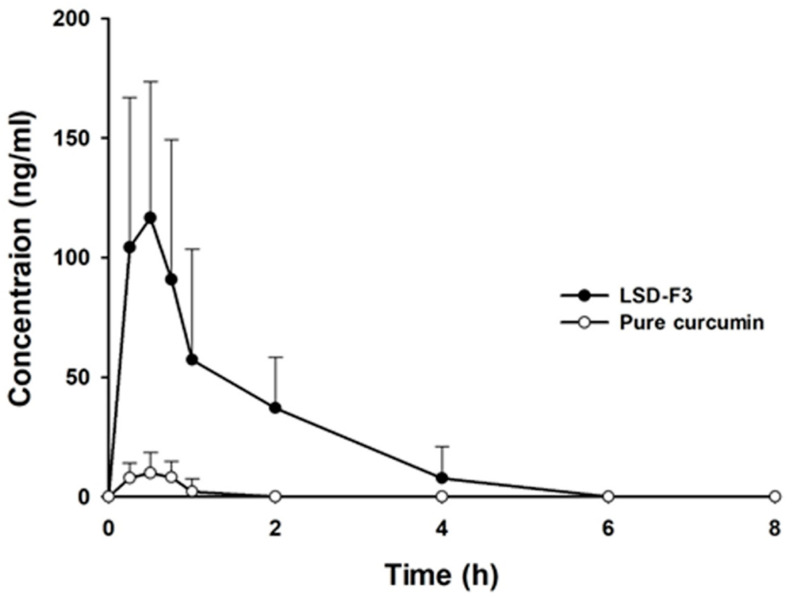
Pharmacokinetic profiles of curcumin following an oral administration of curcumin in different formulations to rats (Mean ± SD, n = 6). The dose was equivalent to 100 mg/kg of curcumin.

**Table 1 pharmaceutics-14-02269-t001:** Composition of the lipid-based solid dispersions.

Formulation	Ratio (*w*/*w*/*w*)			
	Curcumin	Krill Oil	Aminoclay	TPGS (%)
F1	1	1	5	0.5
F2	1	3	5	0.5
F3	1	5	5	0.5
F4	1	5	3	0.5
F5	1	5	1	0.5
F6	1	5	5	0.1
F7	1	5	5	1

**Table 2 pharmaceutics-14-02269-t002:** Storage stability of LSD-F3 at 4 °C and 25 °C (Mean ± SD, n = 3).

Temperature (°C)	Dissolution (%)		
	Day 0	1 Month	2 Month
4	91.9 ± 2.14	93.6 ± 5.62	92.8 ± 3.77
25	91.9 ± 2.14	92.0 ± 5.35	90.1 ± 1.98

**Table 3 pharmaceutics-14-02269-t003:** Pharmacokinetic parameters following an oral administration of curcumin in different formulations to rats (Mean ± S.D, n = 6).

Parameter	Pure Curcumin	LSD-F3
C_max_ (ng/mL)	11.8 ± 7.19	147 ± 49.5 *
T_max_ (h)	0.38 ± 0.26	0.42 ± 0.20
AUC_0–8h_ (ng·h/mL)	7.82 ± 7.02	185 ± 31.7 *

*: *p* < 0.05, statistically significant difference compared to pure curcumin. The dose was equivalent to 100 mg/kg of curcumin.

## Data Availability

All data relevant to the publication are included.

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
