# Peer review of "Lipid/Clay-Based Solid Dispersion Formulation for Improving the Oral Bioavailability of Curcumin"

_pharmaceutics, 2022, doi:10.3390/pharmaceutics14112269_

Round 1

Reviewer 1 Report

A well written paper concerning the “Lipid/clay-based solid dispersion formulation for improving the oral bioavailability of curcumin”.

The authors have presented a new formulation prepared by mixing and consequently lyophilising of a curcumin/krill oil acetonic solution with a surfactant/aminoclay water solution. Several formulations have been tested showing that the optimal weight ratio for the enhancement of curcumin’s dissolution and oral bioavailability was curcumin:krill oil:aminoclay 1:5:5 with 0.5% of D-α-tocopherol polyethylene glycol succinate.

The stability of the formulations was showed by performing experiments at various pH and storage conditions.

Pharmaceutics journal is a suitable journal for this paper, and the recommendation is to publish it pending minor revision.

Manuscript

Raw 96: The title of Table 1 is missing.

Raw 184: It is stated that “Since Omega-3 phospholipids in krill oil could serve as a surfactant…” What do you expect when higher amounts of krill oil are used? (1:6:5)? Can you avoid the use of TPGS?

Raw 220-221: There is a concern regarding the use of Differential Scanning Calorimetry as method for proofing the change of the status of the formulation “from crystalline to amorphous”. As from Figure 2, the endothermic transition at 180 oC attributed to the melting of curcumin disappears when formulated with the other ingredients suggesting the coating of the molecule which is also evident in Figure 4B captured by SEM analysis. Can the authors explain more on how this coating is happening? Is this physical or the curcumin molecules interact with other parts (which) of the formulation? What type of interactions are predicted?

Raw 288: what do you mean with “Amorphous solid dispersion may undergo recrystallization during storage”? Could you check it and explain more? Can be a separation of components instead?

Raw 314 and Raw 299: Tmax for pure curcumin and LSD-F3 are very similar. Could you explain why? How the neat curcumin -not soluble in physiological fluids, shows a fast absorption (0.38 h)?

Reviewer 2 Report

The article by Song, J.G. et al., describing curcumin solid formulation seems scientifically correct and organised, but I see a major issue regarding the scientific rationale of the text:

-    There is a mixture between matherials & methods and results sections, regarding the selection of formulation F3 as the best one. The rationale for such conclusion should be in the results section, and the materials and methods section should avoid to make such conclusions and also avoid the particular attention made to F3 formulation (sections 2.2, 2.3, 2.5…). What I mean is that you should only write in materials and methods all the materials and the methods, avoiding any suggestion that a particular formulation (of the seven examined) is better than the others. This should be explained in the results section and discussed in the discussion section (or presented and discussed if you merge both sections). The abstract should be also modified with this concept in mind.

-    However, I understand that it´s not an easy matter, as long as you quickly detect the best formulation, and then you focus on the best formulation. Please, make an effort in making a scientifically logical text. May be that if you think materials & methods is a secondary section after the references can help you to improve this point.

The text has a generally good style, with some linguistic faults, noted as minor issues.

-    Abstract formulation F3, there are 7 different

-    Introduction, page 2, line 33. Infectious and cardiovascular diseases.

-    Introduction, page 2, line 37. Expanding its… the…

-    Introduction, page 2, line 39. I think there are various lipid-based formulations, as you refer vaguely to “poorly soluble drugs”. Please, use the plural form (“formulations”).

-    Introduction, page 2, line 42. You should broaden a little the concept “first-pass”. May be “hepatic first-pass”?

-    Introduction, page 2, line 42-43. “enhancing oral absorption”.

-    Introduction, page 2, lines 43-44. Among various lipids, krill oil, rich in phospholipids and coupled with long-chain omega-3 polyunsaturated fatty acids, is useful for the preparation of lipid-based formulations.

-    Introduction, page 2, line 52. I think you should include a comma after cryoprotectant to improve the sentence.

-    Introduction, page 3, line 58. Formulations WERE prepared.

-    Materials, page 4. The typesetting of table 1 is not correct. It is in the middle of the paragraph, away from the legend.

-    Results, page 8, line 187. You don´t explain why did you choose a 1:5 ratio. It is described in the following paragraph, may be you should modify and move this sentence below.

-    Results, page 12, Figure 4. The inserted text is readable in the bigger images, but in the smaller image I can´t read it. I would like to see the scale bar, may be a solution is to have the 3 images with the same size. It would be nice if two of the images had the same magnification for comparative purposes…

Reviewer 4 Report

The conduction of the study proved to be adequate, leading to the obtention of interesting and very promising results.

Here are some suggestions so that they can contribute to the quality of the presentation of the manuscript:

1) It would be more appropriate to use the word “DSC curves” instead of “thermograms”;

2) What was kind of syringe filters used? What was the material composition (Nylon, PVDF...) and the diameter;

3) How was the choice of 2% tween in the dissolution medium? Where was the choice based?

4) For the FASSIF and FESSIF assays: Were they run at 50 rpm and 900 mL?

5) About the temperatures used in the Stability test. Why the authors used 4 °C and 25 °C? Wouldn't it be more interesting to leave it at a higher temperature?

6) Review the discussion about Tmax, because fast absorption was not observed. It appears that there are no significant differences between the Tmax from the two formulations.

Round 2

Reviewer 2 Report

It´s now difficult for me to find any flaw in the text by Song, J.G. et al. I like the inclusion of SEM images of the dispersion at different moments to better illustrate the stability.

Reviewer 3 Report

Responses and revisions are acceptable.